# Hyperspectral Ophthalmoscope Images for the Diagnosis of Diabetic Retinopathy Stage

**DOI:** 10.3390/jcm9061613

**Published:** 2020-05-26

**Authors:** Hsin-Yu Yao, Kuang-Wen Tseng, Hong-Thai Nguyen, Chie-Tong Kuo, Hsiang-Chen Wang

**Affiliations:** 1Department of Ophthalmology, Kaohsiung Armed Forced General Hospital, Kaohsiung City 80284, Taiwan; dokiki21@gmail.com; 2Department of Medicine, Mackay Medical College, 46, Sec. 3, Zhongzheng Rd., Sanzhi Dist., New Taipei 25245, Taiwan; tseng@mmc.edu.tw; 3Department of Mechanical Engineering and Center for Innovative Research on Aging Society (CIRAS), National Chung Cheng University, 168, University Rd., Min Hsiung, Chia Yi 62102, Taiwan; g06441054@ccu.edu.tw; 4Department of Optometry and Innovation Incubation Center, Shu-Zen Junior College of Medicine and Management, Kaohsiung 821, Taiwan; ctkuo@mail.nsysu.edu.tw

**Keywords:** diabetic retinopathy stage, hyperspectral imaging, ophthalmoscope, fluorescein angiography, optical coherence tomography

## Abstract

A methodology that applies hyperspectral imaging (HSI) on ophthalmoscope images to identify diabetic retinopathy (DR) stage is demonstrated. First, an algorithm for HSI image analysis is applied to the average reflectance spectra of simulated arteries and veins in ophthalmoscope images. Second, the average simulated spectra are categorized by using a principal component analysis (PCA) score plot. Third, Beer-Lambert law is applied to calculate vessel oxygen saturation in the ophthalmoscope images, and oxygenation maps are obtained. The average reflectance spectra and PCA results indicate that average reflectance changes with the deterioration of DR. The G-channel gradually decreases because of vascular disease, whereas the R-channel gradually increases with oxygen saturation in the vessels. As DR deteriorates, the oxygen utilization of retinal tissues gradually decreases, and thus oxygen saturation in the veins gradually increases. The sensitivity of diagnosis is based on the severity of retinopathy due to diabetes. Normal, background DR (BDR), pre-proliferative DR (PPDR), and proliferative DR (PDR) are arranged in order of 90.00%, 81.13%, 87.75%, and 93.75%, respectively; the accuracy is 90%, 86%, 86%, 90%, respectively. The F1-scores are 90% (Normal), 83.49% (BDR), 86.86% (PPDR), and 91.83% (PDR), and the accuracy rates are 95%, 91.5%, 93.5%, and 96%, respectively.

## 1. Introduction

A high circulating blood glucose is a major feature of diabetes, which can be divided into insulin-dependent diabetes mellitus (type 1 diabetes) and non-insulin-dependent diabetes mellitus (type 2 diabetes) [1,2]. Diabetes heavily burdens society and the economy. According to the IDF data in 2014, approximately four million people died from diabetic complications. Diabetes accounts for 2.5–15% of the annual health care budget. In recent years, significant progress in the education on and treatment of diabetes has been made, but the prevalence of diabetes still has not decreased [2].

Chronic increase in circulating blood glucose can damage small and large blood vessels and results in many complications. Diabetic retinopathy (DR) is one of the major complications of microvascular injury. Patients with early-stage DR are asymptomatic, and DR is often diagnosed at its advanced stages when it already compromises vision. Without timely control, the condition may lead to vision loss [3]. The early clinical features of DR are microvascular and intraretinal hemorrhage [3,4]. More than 95% of patients have a 20-year history of diabetes, and 80% of these patients have type 2 DR. Animal studies have shown that early DR causes the thickening of the basement membrane of the retina and accelerates the apoptosis of microvascular cells [5,6,7]. As DR progresses, the number and area of bleeding increase, and cotton-like spots appear because of blocked microvascular anterior small arteries. Neovascularization initiates because microvessels are not perfused, and proliferative DR (PDR) is prone to retinal detachment if not treated in time and leads to vision loss [3]. DR is the leading cause of blindness in individuals with age of 20–65 years. Approximately 93 million people worldwide suffer from DR, and 17 million of these people have PDR. Additionally, 28 million people suffering from DR lose their sight. A total of 155 million people are expected to suffer from DR in 2030, and more than 51 million people may lose their sight [2]. Therefore, the early detection and treatment of DR are crucial.

The diagnosis of DR mainly is based on retinal changes. A retinal image captured by a fundus camera at a high resolution can display retinal color and blood vessels, macula color, and important structures, such as the optic nerve. Retinal imaging is preferred in diabetic retinal screening. Therefore, retinal images captured by fundus cameras have been extensively used in the detection of the different lesion types of DR by image processing algorithms and in automatic DR detection and classification [8,9,10,11,12]. Abnormal neovascularization is an important clinical feature of PDR. Welikala et al. used two independent vascular segmentation methods for normal blood vessels, neovascularization, and bright lesions [13]. The characteristics of abnormal neovascular types were extracted and then supported by a support vector machine for classification and statistics. The results of the two independent systems were combined for the identification of the lesion areas with abnormal neovascularization in the retinal images.

The hyperspectral imaging (HSI) technology has been widely used to detect DR. V. Nourrit et al., used the standard Zeiss ophthalmoscope with liquid crystal adjustable filter and high-resolution digital camera to obtain fundus images in different bands, and the Beer-Lambert law can be used in the estimation of oxygen concentrations in retinal blood vessels [14]. Amir H. Kashani et al., installed a hyperspectral computed tomographic imaging spectrometer on a standard Zeiss fundus camera to capture fundus images at different wavelengths. The absorbance values of two different wavelengths were separately measured and then calculated by the Beer-Lambert law. Oxygen concentrations in the blood vessels in the fundus image were then obtained [15]. By conducting statistical analysis on arterial and venous oxygen concentrations in the retinal vessels of different types of DR, Amir H. Kashani et al., demonstrated that the difference in oxygen concentration between arteries and veins is significantly lower in patients with PDR than that in patients with normal or other types of DR [15].

Fluorescein angiography (FA) and optical coherence tomography (OCT) are also common methods for detecting DR [16,17,18,19]. FA is preferred in detecting new blood vessels, but it cannot produce three-dimensional information because of its poor axial resolution. Moreover, it is easily hindered by fluorescein leakage from new blood vessels and requires a long measurement time. Standard OCT can provide limited position information in the detection of new blood vessels, and the doppler OCT imaging technology can directly evaluate the three-dimensional information of new blood vessels. Masahiro Miura et al. used standard OCT to detect the location of a proliferating layer and used Doppler OCT to observe the blood flow structure of a neovascular tissue. Doppler OCT imaging can provide a noninvasive and short-term measurement method for detecting PDR [20,21]. In this study, FA and OCT are utilized to diagnose and build a reliable database of different DR stages. Next, we used the principle of HSI and principal component analysis (PCA) to provide a new optical detection method for comparing the spectrum characteristics of blood vessels in the fundus camera images of different types of DR and the results of the principal component scores [22,23]. HSI will use the golden database established by FA and OCT to identify the corresponding DR stage. With the spectral characteristics of this method, we believe that it will enable the rapid identification of DR stage.

## 2. Materials and Methods

### 2.1. Sample Preparation

In this study, the ophthalmology department of Kaohsiung Armed Forces General Hospital provided all the pictures of the pathological sections and the patients. A total of 200 patients were included for the construction of a hyperspectral ophthalmoscope images (HSOI) database. The patients comprised 50 Normal, 50 background DR (BDR), 50 pro-proliferative DR (PPDR), and 50 PDR cases.

### 2.2. Ethical Statement

The Institutional Review Board of the Kaohsiung Armed Forces General Hospital approved the informed consent forms and the study protocol (IRB Number: KAFGHIRB-E(I)-20180271). All participants gave their informed consent. All approaches were carried out in accordance with the regulations and relevant guidelines. The experiment design and experiment process are investigated by the ethics committee of the Kaohsiung Armed Forces General Hospital and not involved ethical experiments.

### 2.3. Clinical Features and Stages of Diabetic Retinopathy

The cause of blindness in approximately 5% of the world’s blind population is DR, which is caused by leakage and bleeding induced by hyperglycemia in small blood vessels that supply the retina [3,24]. Almost half of people with diabetes develop multiple degrees of retinopathy, and individuals with prolonged diabetes have increased risk of acquiring retinopathy. In the early stage of retinopathy, only slight blurring of vision occurs and thus usually ignored. However, the condition gradually deteriorates vision or even lead to blindness; thus, early diagnosis and treatment are necessary to prevent blindness. A fundus camera can observe retinal color, blood vessels, macula color, and important structures, such as the optic nerve. Using a fundus camera is preferred in diabetic retinal screening. However, DR is not easily identified in staging and relies heavily on the subjective judgment of clinicians. In particular, a DR lesion is characterized by BDR and PDR or PPDR. Staging is difficult, but it affects the important basis for future vision recovery or blindness of the patient. The diabetic retinopathy can be classified into background diabetic retinopathy (BDR), pre-proliferative diabetic retinopathy (PPDR), and proliferative diabetic retinopathy (PDR). BDR presents dot-blot or flame-shaped retinal hemorrhages, hard exudates, and microaneurysm in ophthalmoscope. PDR presents same features in ophthalmoscope of BDR plus neovascularization in optic disc or retina. PPDR is a stage between BDR and PDR, and it has same features in ophthalmoscope of BDR plus retinal nerve fiber layer infarction (seen as cotton-wool spot in ophthalmoscope), but no neovascularization in optic disc or retina [25].

### 2.4. Fluorescein Angiography

Figure 1 shows normal, BDR, PPDR, and PDR FA images. Figure 1a shows the normal FA image of the right eye, which shows no sign of disease or pathology. The gaze was into the camera so the macula was in the center of the image and the optic disk was located toward the nose. The optic disk showed pigmentation at the perimeter of its lateral side. This phenomenon is considered normal. In the BDR stage, microaneurysms and retinal punctate hemorrhage were observed. Figure 1b shows the high-fluorescence small early microangiomas, which are highlighted by the red circle. In the late BDR stage, given that the fluorescent agent in the microvessels leaked out, the FA image showed a wide range of high fluorescence. The clinical feature of PPDR is progressive retinal ischemia. Figure 1c shows a wide range of low fluorescence because the retinal microvessels were not dropped out. Neovascularization is an evidence of PDR, and nearly more than a quarter of retinal microvasculature has no perfusion before PDR. Neovascularization manifests as hypervascularization on disk (NVD) or hypervascularization elsewhere (NVE) hyperplasia in the optic nerve head. In this part of the FA, early images can highlight new blood vessels, as shown in Figure 1d in the red circle frame. Late images show a wide range of high fluorescence because stains leaked from new blood vessel tissues, which are outlined by the red circle in Figure 1e. The operational theory of FA is shown in Appendix A.

### 2.5. Ophthalmoscope Images

Figure 2 shows the ophthalmoscope images with clinical features of DR. Figure 2a shows the normal condition with clear macular and microvascular distribution. Figure 2b shows the ophthalmoscope image of a patient with BDR, where the black, blue, and white boxes are the location of the small aneurysm, hard exudate, and hemorrhagic lesions, respectively. Figure 2c–e shows the ophthalmoscope images of a patient with PPDR. The black round frame in Figure 2c–e shows cotton-like spots, abnormal small vessels, and dark stained hemorrhagic lesions, respectively. Figure 2f,g shows the ophthalmoscope images of patients with PDR. The black round frames in Figure 2f,g shows NVD and NVE, respectively. The experimental setup of the ophthalmoscope image measurement system is shown in Appendix A.

### 2.6. Optical Coherence Tomography Images

Figure 3 shows the OCT images with the stages of DR. Figure 3a shows normal and no lesions. The feature outlined by a red circle was selected as the choroidal blood vessel. The black, green, orange, and blue arrows are the vitreous body, photoreceptor cell, retinal inner limiting membrane, and retinal pigment epithelium, respectively. BDR is an early change of DR. The blood and body fluid exudate of the vascular exudation can cause retinal edema or sedimentation, which may cause the deterioration of vision. BDR is taken by OCT, as shown in Figure 3b,c, which shows that microvascular obstruction in PPDR causes the hypoxia and ischemia of the local retina, and the formation of abnormal arteries and veins causes severe leakage and bleeding. Many blood vessels are located in the red circle. Blood vessels that grow on the surface of the retina or optic nerve affected by PDR have fragile walls. When the retina grows, these walls easily rupture and thus leak blood into the vitreous, causing the vitreous to bleed. Intrahemorrhage or pulling out the retinal detachment results in the significant loss of vision or even blindness. The OCT image in Figure 3d indicates that the macular part of the center of the retina is the most concentrated area of photoreceptor cells. In the red circle, the macular area is evidently edematous. The experimental setup of OCT system is shown in Appendix A.

### 2.7. Hyperspectral Ophthalmoscope Images

The estimated spectral processes of the hyperspectral ophthalmoscope imaging (HSOI) data are illustrated in Figure 4. HSOI was used to obtain the spectrum of each image element of an endoscopic image. HSOI was divided into three parts, namely, PCA for spectral data reduction, transformation matrix calculation for determining the relationship between an ophthalmoscope (Kowa, Nonmyd 7, Torrance, CA, USA) and spectrophotometer (Ocean Optics, QE65000, Dunedin, FL, USA), and the spectral reproduction of images. The calculations and experimental outcomes are described in [26,27,28,29,30,31,32]. After capture was corrected, the procedure for processing the red, green, and blue (RGB) values of each image element was conducted for a precise estimation of the spectra. To match the color performance of the spectrophotometer and the camera, correction of color was implemented. The International Commission on Illumination/Commission International de L’éclairage XYZ tristimulus values were studied and used to establish RGB values from the spectra of 24-color checkers (Mini Color Checkers, X-Rite, Grand Rapids, MI, USA) as standard values according to the spectrophotometric data. Under similar lighting conditions, images captured through endoscopy and a computer program were used to retrieve the RGB values of each image element. Finally, third-order polynomial regression was performed for the RGB components separately and determined the color relationship between the two devices. The xenon lamp, in the output format of sRGB (RAW image files) in commercial endoscopy, provided the reference white, which is different from the artificial lights used to measure the spectra of 24-color checkers. Therefore, chromatic adaptation transformation was carried out before third-order polynomial regression.

### 2.8. Hyperspectral Ophthalmoscope Imaging Calculated Processes

The calculation procedure of the average spectra of endoscopic images with HSI data are shown in Figure 4. The HSOI technique was generally applied to acquire the spectrum of each local image element. First, a spectrophotometer (Ocean Optics, QE65000) was applied to measure the spectra of 24 Macbeth color checkers under the illumination of a notable uniform artificial light. In the visible light region (380–780 nm), the reflection spectrum of each color checker was then acquired. These spectra were arrayed as a matrix, that is, ${\left[D\right]}_{401x24}$, where the columns are the numbers of the color checkers and the rows are the intensities of the wavelengths at 1 nm intervals. Six eigenvectors with the most considerable contribution were set as the foundation for spectral calculation and arrayed as a matrix ${\left[E\right]}_{6x401}$ through calculating the eigen system and adopting the principal component analysis. The corresponding eigen values of these six eigenvectors ${\left[\alpha \right]}_{6x24}$ could be decided as below:(1)$${\left[\alpha \right]}^{T}={\left[D\right]}^{T}pinv\left[E\right],$$
where *pinν* indicates the pseudoinverse of the matrix. Under the same illumination condition, a digital camera captured the color checkers, and the output image file format was sRGB (JPEG image files). Computer programs were used to obtain the red, green, and blue values (0–255) of each color checker’s image and were subsequently marked on a range of [R]_srgb_, [G]_srgb_, and [B]_srgb_ (0–1). CIE XYZ tristimulus values could be obtained by transforming these RGB values through the formula mentioned below:(2)$$\left[\begin{array}{c} X\\ Y\\ Z \end{array}\right]=\left[T\right]\left[\begin{array}{c} f\left(R_{srgb}\right)\\ f\left(G_{srgb}\right)\\ f\left(B_{srgb}\right) \end{array}\right],$$
where
(3)$$\left[T\right]=\left[\begin{array}{ccc} 0.4124 & 0.3576 & 0.1805\\ 0.2126 & 0.7152 & 0.0722\\ 0.0193 & 0.1192 & 0.9505 \end{array}\right],$$
(4)$$f\left(n\right)=\left\{\begin{array}{c} {\left(\frac{n+0.055}{1.055}\right)}^{2.4}\;\;,\;n>0.045\\ \left(\frac{n}{12.92}\right)\;\;\;\;\;\;\;\;\;\;\;,otherwise \end{array}\right..$$

Given that a CIE standard light source D65 illuminated the reference white of the sRGB color space, by applying CMCCAT2000, the RGB values were calibrated for chromatic adaptation; such standard light source was different from the artificial light used to measure the spectra of color checkers. Given the accuracy of spectral calculation, the color calibration of the camera was needed. According to Equations (5)–(7), CIE XYZ tristimulus values could be obtained by transforming the reflection spectra determined by the spectrophotometer. Among these calculations, $\bar{x}\left(\lambda \right)$, $\bar{y}\left(\lambda \right)$, and$\;\bar{z}\left(\lambda \right)\;$are the color matching functions, R(λ) is the spectral reflectance of the respective color checker, and S(λ) is the relative spectral power distribution of the artificial light.
(5)$$X=k\int_{380nm}^{780nm}S\left(\lambda \right)R\left(\lambda \right)\bar{x}\left(\lambda \right)d\lambda ,$$
(6)$$Y=k\int_{380nm}^{780nm}S\left(\lambda \right)R\left(\lambda \right)\bar{y}\left(\lambda \right)d\lambda ,$$
(7)$$Z=k\int_{380nm}^{780nm}S\left(\lambda \right)R\left(\lambda \right)\bar{z}\left(\lambda \right)d\lambda ,$$
where
(8)$$k=100/\int_{380nm}^{780nm}S\left(\lambda \right)R\left(\lambda \right)\bar{y}\left(\lambda \right)d\lambda .$$

When the chromatic adaptation was transformed, the RGB values matching to the new XYZ values were estimated by the reverse processes of Equations (2)–(4) and considered standard matrix $\left[A\right]$. Through using the third-order polynomial regression for the red, green, and blue elements separately, the color correlation between the spectrophotometer and camera was obtained; the regression matrix $\left[C\right]\;$is defined as follows:(9)$$\left[C\right]=\left[A\right]pinv\left[F\right],$$
where
(10)$$\left[F\right]={\left[\begin{array}{c} 1,R,G,B,\mathrm{RG},\mathrm{GB},\mathrm{BR},R^{2},G^{2},B^{2},\mathrm{RGB},R^{3}\\ ,G^{3},B^{3},{\mathrm{RG}}^{2},{\mathrm{RB}}^{2},GR^{2},GB^{2},BR^{2},BG^{2} \end{array}\right]}^{T},$$
where R, G, and B are the camera capturing the RGB values from color checkers. In this study, [K] represents the RGB values captured from any of the images that expands into a format, such as the original matrix [F], and arrayed as a matrix [β].
(11)$$\left[\mathrm{Corrected}\;\mathrm{RGB}\right]=\left[C\right]\left[K\right].$$

Eventually, a transformation matrix $\left[M\right]$ among the camera and spectrophotometer was determined by the following equation:(12)$$\left[M\right]=\left[\alpha \right]pinv\left[\beta \right].$$

The RGB values were multiplied by the regression matrix [C] for each pixel in any one of the images captured by the camera. With the help of Equations (2)-(4), the corresponding XYZ values could be estimated. The calculated spectra in the visible light range (380–780 nm) were determined according to the following equation:(13)$${\left[Spectra\right]}_{380-780nm}=\left[E\right]\left[M\right]\left[\begin{array}{c} X\\ Y\\ Z \end{array}\right].$$

### 2.9. Retinal Image Processing Algorithm

To automate the retrieval of vascular information in the retinal images, we used a new image processing algorithm, as shown in Figure 5. We divided the calculation process into the following five steps: preprocessing, vascular enhancement, contrast enhancement, binarization, and masking. In the preprocessing step, the color of the blood vessels was darker than that of the lesions, such as microvessels and hemorrhage. These regions must be excluded during blood vessel detection for the prevention of misjudgment [33,34,35]. In the vascular enhancement step, a two-dimensional Gabor filter enables optimal localization in spatial and frequency domains. Similar to human biovisual characteristics, it can simply describe the spatial frequency, spatial position, and direction selectivity of an image [33,35,36]. In contrast enhancement, the brightness value of an image-specific interval to a new interval value was mapped, and the mapped curve shape was specified on the basis of a gamma parameter. In the binarization step, the threshold of the image was determined through Otsu’s binarization method so that the between-class variance had the maximum value [37]. Finally, we multiplied the image by a mask to hide unwanted edge information. Through the above steps, we could automatically capture retinal vascular information. The detailed explanation of this algorithm is reported in Appendix A.

### 2.10. Blood Oxygen Saturation Calculation

Beer’s law, also known as the Beer-Lambert law, is an optical fundamental law. When light penetrates a sample solution, light absorbance (*A*) is proportional to absorption coefficient (*ε*), optical path length (*d*), and concentration (*c*). Absorption coefficient (absorptivity or absorption coefficient) is also known as the extinction coefficient (*k*). The optical path length *d* of a sample solution is expressed in centimeters, and *c* is expressed in molar concentration. Absorption coefficient is multiplied by the reciprocal of centimeter to the reciprocal of molar concentration (cm^−1^M^−1^) as a unit. The absorption coefficient at this time can then be called molar absorptivity, and its symbol is often represented by *ε*. The common Beer’s law is expressed as:(14)A=εdc.

Another important definition we must understand in Equation (14) is absorbance of light (absorbance). When light hits a sample solution, some of the light is absorbed by the sample solution, and the remaining light penetrates the sample solution. That is, light transmittance *T* (transmittance) or ratio of light penetration can be obtained by dividing original light incident ray intensity I_0_ by transmitted light intensity, which becomes I_1_, and then multiplying the result by 100 as follows:(15)$$T=\frac{I_{1}}{I_{0}}\times 100\%.$$

Conversely, some of the light is absorbed by the sample, and the absorbance *A* of the defined light is:(16)$$A=-logT=\log\left(\frac{1}{T}\right).$$

In this study, we applied the Beer-Lambert law to calculate oxygen saturation in the blood vessels in the retinal image. The absorbance of the blood vessel at wavelength λ is:(17)$$A\left(\lambda \right)=c_{1}d\varepsilon_{HbO}\lambda +c_{2}d\varepsilon_{Hb}\lambda ,$$
where $\varepsilon_{HbO^{\lambda }}$ and $\varepsilon_{Hb^{\lambda }}$ are the molar absorption coefficients of oxygenated hemoglobin and no oxygenated hemoglobin at wavelength *λ*, respectively, and *d* is blood vessel width.

In Equation (17), A(λ) was obtained by the hyperspectral algorithm. We could arbitrarily select two wavelengths $\lambda_{1},{\;\lambda }_{2\;}$to determine $A_{1}\left(\lambda_{1}\right)$ and $A_{2}\left(\lambda_{2}\right)$. The molecular absorption coefficient spectra of oxygenated and non-oxygenated hemoglobin are indicated by Prahl’s work. We could obtain the values of $\varepsilon_{{\mathrm{HbO}}^{\lambda_{1}}}$, $\varepsilon_{{\mathrm{HbO}}^{\lambda_{2}}}$, $\varepsilon_{{\mathrm{Hb}}^{\lambda_{1}}}$, and $\varepsilon_{{\mathrm{Hb}}^{\lambda_{2}}}$. We used the average between the thickness of retinal vascular arteries (0.104 mm) and that of veins (0.1325 mm) as the thickness of the blood vessel in the width of the blood vessel. Finally, the oxygenated hemoglobin concentration c_1_ and the oxygen-free hemoglobin concentration c_2_ of the blood vessel could be calculated.

## 3. Results

### 3.1. Average Reflection Spectrum

The average reflectance spectrum of the arteries and veins at different DR stages can be obtained by HSI. Figure 6a shows the mean reflectance spectrum of different DR stages. The difference in reflectance spectrum shows a decreasing spectral reflectance according to the degree of lesion (Normal, BDR, PPDR, and PDR). In general, diabetes causes retinal artery disease. Blood vessels gradually lose their pericytes and basement membranes, and endothelial cells present damage and proliferation because of vascular occlusion [25]. As a result, spectral reflectivity decreases as the staging becomes severe in the green band (495–570 nm). The reflectivity of PDR in the red band (620–780 nm) is significantly reduced. Oxygen saturation in the retinal arteries decreases with the deterioration of DR. In the red band, absorption is lower in the presence of oxygenated hemoglobin than that observed in the presence of non-oxygenated hemoglobin [38]. Therefore, as the oxygen content in the blood vessels decreases, the proportion of anaerobic hemoglobin with enhanced light absorption increases, resulting in a decrease in spectral reflectance in the red band. Figure 6b shows the average reflectance spectra of DR veins at different stages. The overall intensity of the spectral reflectance is weaker than that of the arteries because of thick veins and dark colors. The oxygen saturation of retinal veins increases with the degree of DR and is particularly significant in PDR staging [15]. Therefore, as the degree of lesions increases in the red band, spectral reflectance gradually increases, especially in the PDR stage. In this study we use a relatively simple model to analyze the reflectance spectrum obtained by HSI, but it can be used as a basis for doctors to quickly diagnose diabetic retinopathy stage and pave the way for a more intricate model [39].

### 3.2. Spectral Characteristics

The spectral characteristics of arteries and veins in retinal images obtained by principal component analysis were obtained. Figure 6c shows the spectral characteristics of the arteries in the retinal image. The normal DR is located between the first principal component (FPC) −2.00 and 2.50, and the second principal component (SPC) is between −0.50–0.00; BDR is approximately –2.00 < FPC < 2.50, −0.30 < SPC < 0.10; PPDR is approximately −1.50 < FPC < 2.00, −0.10 < SPC < 0.30; and PDR is approximately −2.00 < FPC < 2.00, −0.10 < SPC < 0.40. However, the spectrum characteristics of the arteries in the normal retinal images have a tendency to diverge, but spectral variability can still be seen in general, with a bottom-up trend. The spectral characteristics of the veins in the fundus image are shown in Figure 6d. The normal range falls to approximately −1.50 < FPC < 2.50, 0.00 < SPC < 0.40; the range of BDR falls to −2.00 < FPC < 2.25, −0.10 < SPC < 0.20; PPDR range falls to −1.25 < FPC < 1.00, −0.15 < SPC < 0.75; PDR range falls to −0.75 < FPC < 2.75, −0.25 < SPC < −0.05. However, a large overlap between BDR and PPDR exists, and we consider the overlap as a fuzzy zone. The spectral characteristics of veins in normal retinal images are similarly divergent, but the spectral characteristics are still roughly different. The trend of spectral characteristics difference is from top to bottom. Refer to our previous work for a detailed calculation of PCA [40].

### 3.3. Oxygen Saturation Profile

Figure 7a–d shows the local retinal images of the Normal, BDR, PPDR, and PDR, and the white marks are the arteries and veins. We extracted each pixel of the arterial and venous vessels in four images and simulated the reflection spectrum of each pixel by using HSI. Through the Beer–Lambert law, we calculated the oxygen saturation of each pixel. We obtained the oxygenation maps of the four images by re-coloring each pixel according to different oxygen saturations. As shown in Figure 7e–h, the oxygen saturation profiles of Normal, BDR, PPDR, and PDR are shown. In the oxygen saturation profile, no evident variation in oxygen saturation is observed in the arteries. In the veins parts, profiles are similar to the trend of the reflection spectrum. Under the PDR stage, oxygen saturation is significantly improved compared with other stages. The retinal arteriovenous occlusion of PDR staging patients can be seen in the large areas of retinal microvascular perfusion, resulting in reduced retinal oxygen utilization [41,42,43,44]. Diseased retina tissues decrease oxygen use. These effects are the main reasons for the significant increase in venous oxygen saturation in PDR.

### 3.4. Patient Referral Decision

By comparing the spectral characteristics of the arteries in the retinal image with the oxygen saturation profile, we can determine whether HSOI can diagnose DR. Table 1 shows the results of the diagnosis of 200 test subjects by using HSOI. Of these patients, 50 are Normal, 53 are BDR, 49 are PPDR, and 48 are PDR. The diagnostic results are displayed in a confusion matrix; 45 of the 50 normal testers were accurately tested, and five were mistakenly judged as BDR; 43 of the 53 BDR testers were accurately tested, and five were misjudged as Normal, four were misjudged as PPDR, and one was misjudged as PDR; 43 of the tested subjects of PPDR were accurately tested, two were misjudged as BDR, and four wwere misjudged as PDR; 45 out of 48 testers were accurately tested, and three were mistakenly judged as PPDR. Table 2 shows the values of sensitivity, precision, F1-score, and accuracy of the test results. The sensitivity of diagnosis is based on the severity of DR. Normal, BDR, PPDR, and PDR are arranged in the order of 90.00%, 81.13%, 87.75%, and 93.75% with precision of 90%, 86%, 86%, 90%, respectively. Their F1-scores are 90%, 83.49%, 86.86%, and 91.83% with accuracies of 95%, 91.5%, 93.5%, and 96%, respectively. The detailed explanation of sensitivity, precision, F1-score, and accuracy is shown in Appendix A.

## 4. Discussion

The result of the evaluation of BDR is the worst because BDR belongs to the earliest DR and thus the least easy to identify. The evaluation result of PDR is the best. DR at this stage is easy to identify, and the doctor does not need auxiliary use. This study is more meaningful that the normal case also has good precision.

The results indicate that two main factors affect the arterial reflectance spectra of retinal images. In the green band (495–570 nm), the thickness of the outer cells and basement membranes of the blood vessels gradually decreases, and endothelial cells are damaged and proliferate because of the occlusion of blood vessels. Therefore, the spectral reflectance trend is caused by the deterioration of the stage. Second, in the red band (620–780 nm), the light absorption of oxygenated hemoglobin is lower than that of non-oxygenated hemoglobin. Therefore, as the oxygen content in the blood vessel decreases, light absorption improves relative to that of oxygenated hemoglobin. The red band shows a decline in spectral reflectance. In the part of the venous reflex spectrum of the retinal image, the oxygen saturation of the retinal vein increases as the degree of DR deteriorates. Therefore, as the degree of lesions increases in the red band, spectral reflectance gradually increases, especially in the PDR stage. In the principal component score map, we found that the principal component analysis can clearly distinguish the spectral features of Normal, BDR, PPDR, and PDR retinal arteries and veins. The spectral characteristics of the arteries can be seen as spectral differences, which have a bottom-up trend according to the stage. However, the spectral characteristics of the veins may be blurred in the BDR and PPDR. Therefore, if the difference is directly identified by the reflection spectrum, the part of the retinal artery can be sampled. In the oxygen saturation profiles of Normal, BDR, PPDR, and PDR, no evident differences exist in the arteries. In the vein part, the oxygen saturation in the PDR stage is significantly higher than in other stages.

In clinical scenario, BDR and PPDR have less impact to the vision in diabetic patients than PDR. In most circumstance, the clinical treatment of BDR and PPDR is observation and follow-up. Nonetheless, PDR poses greater risk of visual impairment and even blindness in lifelong span. The treatment of PDR is more aggressive, includes retinal photocoagulation, intra-vitreous injection of anti-vascular endothelial growth factor (anti-VEGF) [45,46], and retinal surgery. Even undergoing the laser or surgical intervention, the visual prognosis of PDR is worse than BDR and PPDR. The retinal image analysis in the presenting study had an advantage in identifying PDR cases, and it is assumed to provide better clinical benefit for the practitioners. Furthermore, staging of diabetic retinopathy relies solely on clinical discretion of the ophthalmologists in ophthalmoscope examination, and there may be some differences of judgement between each ophthalmologist. The presenting retinal image analysis decreases such subjective and individual differences in a clinical setting.

The macula is located in the center of the retina and is the most sensitive part of vision. It is responsible for central vision and has the function of distinguishing the clarity and color of objects. When the retinal cells are injured by stimulation, an inflammatory response is initiated and the retinal pigment epithelium cells secrete VEGF to promote retinal recovery. VEGF has two major effects, it can both increase vascular permeability and promote angiogenesis. However, if the retina is chronically stimulated, VEGF will be excessively secreted, which not only cause the chronic inflammatory of increased vascular permeability but also promotes the formation of choroidal neovascularization, resulting in macular edema. The invention of anti-VEGF offered a glimmer of hope to some retinal diseases that could not be effectively treated in the past. The purpose of intravitreal injection, for macular lesions, is to shrink choroidal neovascularization and eliminate macular edema and bleeding. As far as diabetic retinopathy is concerned, the main cause is microvascular ischemia and hypoxia caused by diabetes, the retina is exposed to excessive VEGF for a long time, resulting in a series of microvascular diseases, increased vascular permeability and neovascularization. Anti-VEGF treatment can reduce the sustained damage of VEGF2 to retinal microvascular, so it can reduce the severity of the disease. Intravitreal drug delivery treatment is a treatment that directly injects drug into the vitreous body and does not need to be delivered through external cardiovascular system, it is the most direct and effective method in theory. There are currently four main intravitreal injection drugs, three of which are anti-angiogenesis factors: Avastin, Lucentis, Eylea, and the other one is long-acting steroid: Ozurdex. Avastin was originally used for indications such as metastatic colorectal cancer, metastatic breast cancer, and non-small cell lung cancer. It has been off-label used in the treatment of macular lesions. Lucentis is the first intravitreal drug approved by Taiwan National Health Insurance to treat macular lesions. Figure 8 is the fundoscopic image of PDR patients before and after Lucentis treatments. In Figure 8a,b, we can find that the patient is diagnosed with proliferative diabetic retinopathy in both eyes. Fundus images of both eyes show typical lesions such as dot-blot retinal hemorrhage, flame-shaped retinal hemorrhage, hard exudate, microaneurysm, cotton-wool spot, and retinal neovascularization. After 5 injections of Lucentis, the ophthalmoscope image is shown in Figure 8c,d. In Figure 8c,d, we can find that the number of patients with retinal hemorrhage, hard exudate, microaneurysm, cotton-wool spot, and retinal neovascularization are all reduced. Figure 8e,f are the ophthalmoscope images of the left and right eyes after seven injections of Lucentis, respectively. In Figure 8e,f, we can find that the patient’s fundus lesions continue to decrease. Figure 8g shows the mean reflectance spectrum of retinal artery at different DR conditions. The difference in reflectance spectrum shows a decreasing spectral reflectance according to the degree of treatments (Normal, seven Lucentis treatments, five Lucentis treatments, and PDR). Compared with the results of Figure 6a, the spectrum information obtained by HSI can verify the anti-VEGF treatment results. Figure 8h shows the mean reflectance spectrum of retinal vein at different DR conditions. Compared with the results of Figure 6b, the spectrum information obtained through HSI can also verify the anti-VEGF treatment results. In the future, we can use such HSI technology to establish a system to verify the results of anti-VEGF treatment.

## 5. Conclusions

In this study, we demonstrated the use of hyperspectral imaging for identifying various stages of diabetic retinopathy. We translated the concept from endoscopic imaging to ophthalmic images recorded with a standard fundus camera. In total, images from 200 subjects (50 healthy, 150 with diabetic retinopathy) were evaluated. From the color fundus images, this method derived absorption spectra from vessels that were then used to determine the oxygenation level of arteries and veins. The novelty lies on hyperspectral imaging and on the corresponding use of oxygenation levels to determine the stage of diabetic retinopathy. We believe that the proposed method is effective and rapid in early-stage treatment. Depending on the present results, the identification of early DR by hyperspectral image technology achieves rapid diagnosis. Physicians can select suspicious areas in ophthalmoscope images to identify DR at its early stages. The spectral information of each pixel in the selected area can be estimated. The degree of a disease degree in a selection area can be defined through the HSOI database. Through the color markers of the selection area in a monitor, the physician is able to identify DR at its early stages.

## Figures and Tables

**Figure 1 jcm-09-01613-f001:**
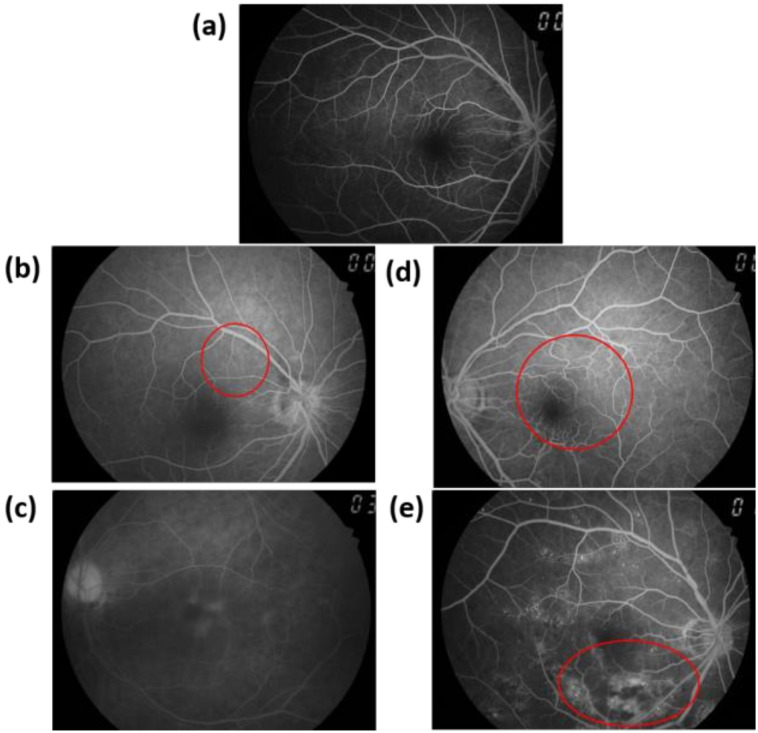
Fluorescein angiography (FA) images of (**a**) Normal, (**b**) background diabetic retinopathy (BDR), (**c**) pre-proliferative diabetic retinopathy (PPDR), and (**d**) early red proliferative diabetic retinopathy (PDR) highlights neovascularization at the red round frame (**e**). The red circle frame is late PDR because it shows a stain that leaks from the new vascular tissue and is shown as high fluorescence.

**Figure 2 jcm-09-01613-f002:**
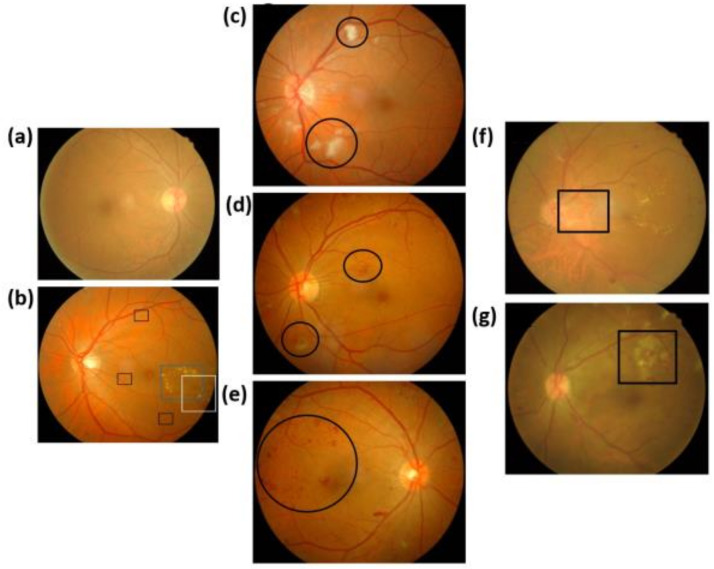
Ophthalmoscope images of (**a**) Normal, (**b**) BDR, (**c**) PPDR with cotton-wool spots, (**d**) PPDR with intraretinal microvascular abnormalities, (**e**) PPDR with dark spotting, (**f**) PDR with neovascularization on disk, and (**g**) PDR with neovascularization elsewhere.

**Figure 3 jcm-09-01613-f003:**
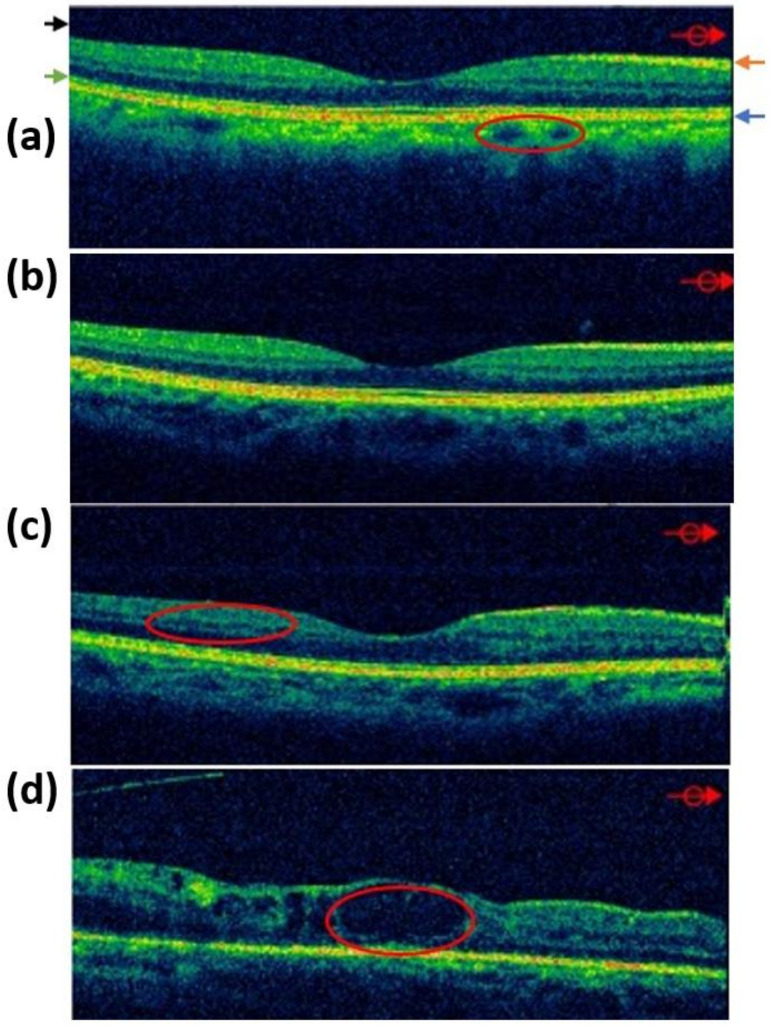
Optical coherence tomography (OCT) images of (**a**) Normal, (**b**) BDR, (**c**) PPDR, and (**d**) PPDR.

**Figure 4 jcm-09-01613-f004:**
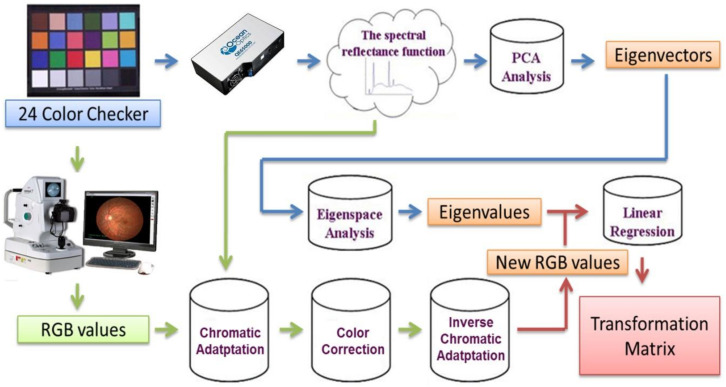
Schematic diagram of the proposed method, using an ophthalmoscope to estimate the spectral reflectance of each pixel in an image.

**Figure 5 jcm-09-01613-f005:**
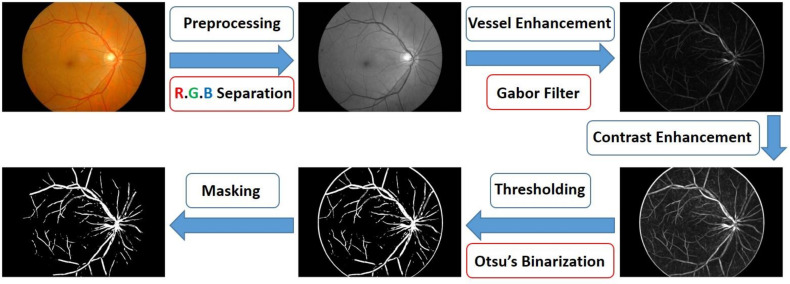
Schematic diagram of the proposed method used in estimating the spectral reflectance of each pixel in an image using an ophthalmoscope.

**Figure 6 jcm-09-01613-f006:**
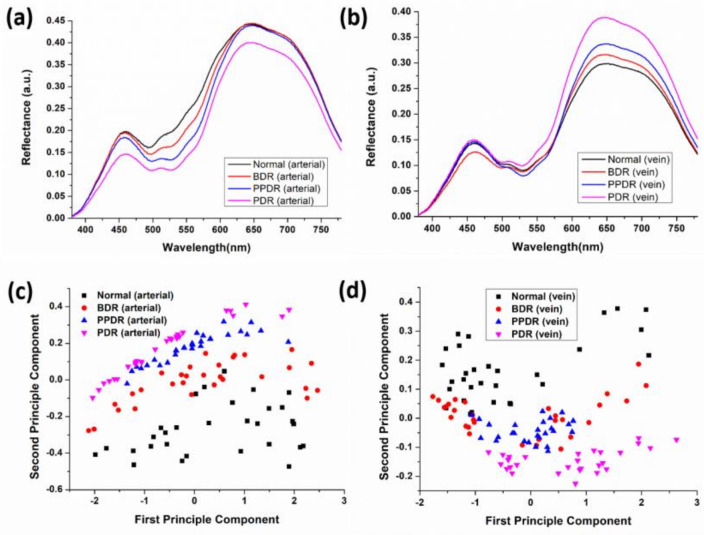
Average reflection spectrum of Normal, BDR, PPDR, and PDR retinas of (**a**) arterial and (**b**) vein; principal component scores of spectral features of (**c**) arterial and (**d**) vein.

**Figure 7 jcm-09-01613-f007:**
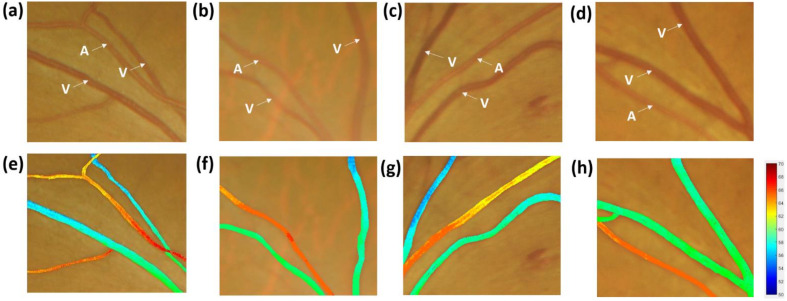
The local retinal images are (**a**) Normal, (**b**) BDR, (**c**) PPDR, and (**d**) PDR; the oxygen saturation profiles are (**e**) Normal, (**f**) BDR, (**g**) PPDR, and (**h**) PDR.

**Figure 8 jcm-09-01613-f008:**
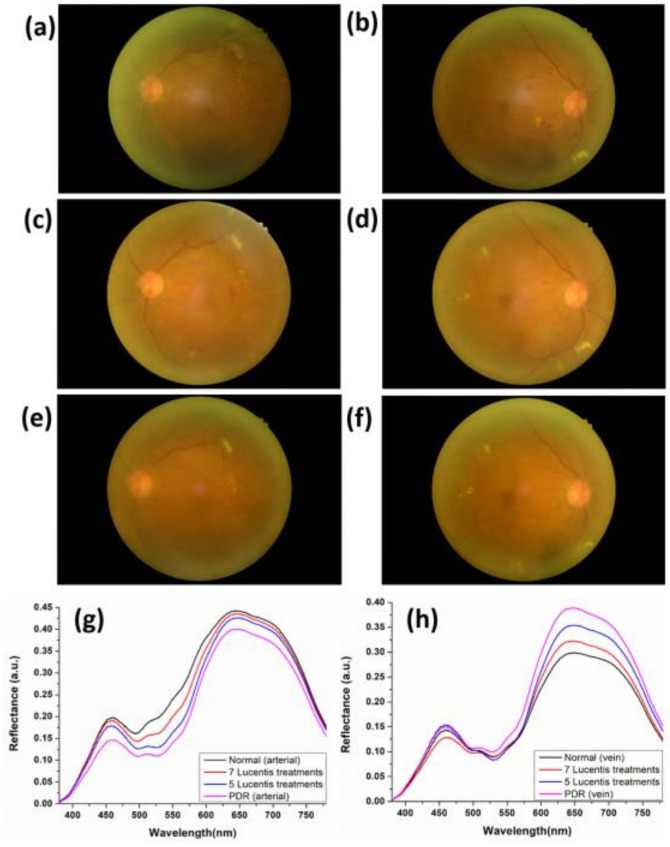
Ophthalmoscope image of PDR patients before and after anti-VEGF treatment: (**a**) Left eye before treatment, (**b**) right eye before treatment, (**c**) left eye after five Lucentis treatments, (**d**) right eye after five Lucentis treatments, (**e**) left eye after seven Lucentis treatments, (**f**) right eye after seven Lucentis treatments; average reflection spectrum of Normal and PDR retinas after five Lucentis and seven Lucentis treatments of (**g**) arterial and (**h**) vein.

**Table 1 jcm-09-01613-t001:** Confusion matrix for 200 test subjects.

	Predicted Referral			
Gold Standard Referral	Normal	BRD	PPDR	PDR
Normal	45	5	0	0
BDR	5	43	4	1
PPDR	0	2	43	4
PDR	0	0	3	45

**Table 2 jcm-09-01613-t002:** Sensitivity, precision, F1-score, and accuracy for 200 test subjects.

	Sensitivity (%)	Precision (%)	F1-Score (%)	Accuracy (%)
Normal	90.00	90.00	90.00	95.00
BDR	81.13	86.00	83.49	91.50
PPDR	87.75	86.00	86.86	93.50
PDR	93.75	90.00	91.83	96.00

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
