# Peer review of "Hyperspectral Ophthalmoscope Images for the Diagnosis of Diabetic Retinopathy Stage"

_jcm, 2020, doi:10.3390/jcm9061613_

Round 1

Reviewer 1 Report

The first three paragraphs in the introduction section can be reduced.

It will be informative if the authors can provide some details and some data if they have on anti-VEGF treated retinal images, that will be really interesting for the readers. 

The authors can increase their discussion section.

Author Response

Reviewer 1:
1. The first three paragraphs in the introduction section can be reduced.
Reply: We cancel the first three paragraphs in the introduction section.
2. It will be informative if the authors can provide some details and some data if they have on anti-VEGF treated retinal images, that will be really interesting for the readers.
Reply: We add six anti-VEGF treated retinal images and descriptions in the discussion section. We also analysis the spectral information of retinal arteries and venous vessels. The added pictures and text descriptions are as follows:
Figure 8. Ophthalmoscope image of PDR patients before and after anti-VEGF treatment,(a)Left eye before treatment (b) Right eye before treatment (c) Left eye after 5 times Lucentis treatments (d) Right eye after 5 times Lucentis treatments (e) Left eye after 7 times Lucentis treatments (f) Right eye after 7 times Lucentis treatments; Average reflection spectrum of Normal, PDR, after 5 times Lucentis, and 7 times Lucentis treatments retinas (g) Arterial (h) Vein.
The macula is located in the center of the retina and is the most sensitive part of vision. It is responsible for central vision and has the function of distinguishing the clarity and color of objects. When the retinal cells are injured by stimulation, an inflammatory response is initiated and the retinal pigment epithelium cells secrete VEGF to promote retinal recovery. VEGF has two major effects, it can both increase vascular permeability and promote angiogenesis. However, if the retina is chronically stimulated, VEGF will be excessively secreted, which not only cause the chronic inflammatory of increased vascular permeability but also promotes the formation of choroidal neovascularization, resulting in macular edema. The invention of Anti-VEGF offered a glimmer of hope to some retinal diseases that could not be effectively treated in the past. The purpose of intravitreal injection, for macular lesions, is to shrink choroidal neovascularization, eliminate macular edema and bleeding. As far as diabetic retinopathy is concerned, the main cause is microvascular ischemia and hypoxia caused by diabetes, the retina is exposed to excessive VEGF for a long time, resulting in a series of microvascular diseases, increased vascular permeability and neovascularization. Anti-VEGF treatment can reduce the sustained damage of VEGF2 to retinal microvascular, so it can reduce the severity of the disease. Intravitreal drug delivery treatment is a treatment that directly injects drug into the vitreous body and does not need to be delivered through external cardiovascular system, it is the most direct and effective method in theory. There are currently four main intravitreal injection drugs, three of which are anti-angiogenesis factors: Avastin, Lucentis, Eylea, and the other one is long-acting steroid: Ozurdex. Avastin was originally used for indications such as metastatic colorectal cancer, metastatic breast cancer, and non-small cell lung cancer. It has been off-label used in the treatment of macular lesions. Lucentis is the first intravitreal drug approved by Taiwan National Health Insurance to treat macular lesions. Figure 8 is the fundoscopic image of PDR patients before and after Lucentis treatments. In Figure 8 (a) and (b), we can find that the patient is diagnosed with proliferative diabetic retinopathy in both eyes. Fundus images of both eyes show typical lesions such as dot-blot retinal hemorrhage, flame-shaped retinal hemorrhage, hard exudate, microaneurysm, cotton-wool spot, and retinal neovascularization. After 5 injections of Lucentis, the ophthalmoscope image is shown in Figure 8 (c) and (d). In Figure 8 (c) and (d), we can find that the number of patients with retinal hemorrhage, hard exudate, microaneurysm, cotton-wool spot, and retinal neovascularization are all reduced. Figures 8 (e) and (f) are the ophthalmoscope images of the left and right eyes after 8 injections of Lucentis, respectively. In Figure 8 (e), (f), we can find that the patient's fundus lesions continue to decrease. Figure 8(g) shows the mean reflectance spectrum of retinal artery at different DR conditions. The difference in reflectance spectrum shows a decreasing spectral reflectance according to the degree of treatments (Normal, 7 Lucentis treatments, 5 Lucentis treatments, and PDR). Compared with the results of Figure 6 (a), the spectrum information obtained by HSI can verify the anti-
VEGF treatment results. Figure 8(h) shows the mean reflectance spectrum of retinal vein at different DR conditions. Compared with the results of Figure 6 (b), the spectrum information obtained through HSI can also verify the anti-VEGF treatment results. In the future, we can use such HSI technology to establish a system to verify the results of anti-VEGF treatment.
3. The authors can increase their discussion section.
Reply: We add some sentences as “In clinical scenario, BDR and Pre-PDR have less impact to the vision in diabetic patients than PDR. In most circumstance, the clinical treatment of BDR and Pre-PDR is observation and follow-up. Nonetheless, PDR poses greater risk of visual impairment and even blindness in lifelong span. The treatment of PDR is more aggressive, includes retinal photocoagulation, intra-vitreous injection of anti-vascular endothelial growth factor (anti-VEGF) [45, 46], and retinal surgery. Even undergoing the laser or surgical intervention, the visual prognosis of PDR is worse than BDR and Pre-PDR. The retinal image analysis in the presenting study had advantage in identifying PDR cases, and it is assumed to provide better clinical benefit for the practitioners. Furthermore, staging of diabetic retinopathy relies solely on clinical discretion of the ophthalmologists in ophthalmoscope examination, and there may be some differences of judgement between each opathalmologists. The presenting retinal image analysis decreases such subjective and individual differences in clinical setting.” in the end of discussion section. We also add two new references as: [45] Ng, E., Shima, D., Calias, P. et al. Pegaptanib, a targeted anti-VEGF aptamer for ocular vascular disease. Nat Rev Drug Discov 5, 123–132 (2006). [46] Simó, R., Hernández, C. Intravitreous anti-VEGF for diabetic retinopathy: hopes and fears for a new therapeutic strategy. Diabetologia 51, 1574 (2008).

Reviewer 2 Report

This interesting study described a hyperspectral imaging for identifying the severity of DR and translated the concept from endoscopic imaging to ophthalmic images recorded with a standard fundus camera. The findings are novel and have practical applicability to retinal practices. 

  1. Which criteria you choose to define the stage of DR?  What is the difference between BDR and PPDR? Page 3 Line 115  "Clinical Features and Stages of Diabetic Retinopathy" can't adequately describe the DR classification.
  2. Age, sex, diopter, glaucoma and other retinal disease may affect retinal oxygen vessel saturation. Does the four groups in the present study control these variables

Author Response

Reviewer 2:
This interesting study described a hyperspectral imaging for identifying the severity of DR and translated the concept from endoscopic imaging to ophthalmic images recorded with a standard fundus camera. The findings are novel and have practical applicability to retinal practices.
1. Which criteria you choose to define the stage of DR?
Reply: The definition of the stage of DR was come from the clinical features with the fluorescein angiography, fundus camera, and OCT observations. We also referenced a book “Bowling, B. Kanski's clinical ophthalmology: a systematic approach; Saunders Ltd: 2015” in chapter 14: Retinal Vascular Disease as reference 36 in the manuscript.
We had described more detail about clinical features of each DR stage in Section 2.
2. What is the difference between BDR and PPDR?
Reply: In the BDR stage, microaneurysms and retinal punctate hemorrhage were observed. Figure. 1(b) shows the high fluorescence small early microangiomas, which are highlighted by the red circle. In the late BDR stage, given that the fluorescent agent in the microvessels leaked out, the FA image showed a wide range of high fluorescence.
The clinical feature of PPDR is progressive retinal ischemia. Figure. 1(c) shows a wide range of low fluorescence because the retinal microvessels were not dropped out. For ophthalmoscope images, Figure. 2(b) shows the ophthalmoscope image of a patient with BDR, where the black, blue, and white boxes are the location of the small
aneurysm, hard exudate, and hemorrhagic lesions, respectively. Figure. 2(c) c)––(e) show the ophthalmoscope images of a patient with P PDR. The black round frame in Figure. 2(c) c)––(e) shows cotton like spots, abnormal small vessels, and dark stained hemorrhagic lesions, respectively. In optical coherence tomography images, BDR is an early change of DR. The blood and body fluid exudate of t he vascular exudation can cause retinal edema or
sedimentation, which may cause the deterioration of vision. BDR is taken by OCT, as shown in Figure. 3(b). Figure. 3(c) shows that microvascular obstruction in PPDR causes the hypoxia and ischemia of the loc al retina, and the formation of abnormal arteries and veins causes severe leakage and bleeding.
3. Page 3 Line 115 "Clinical Features and Stages of Diabetic Retinopathy" can't adequately describe the DR classification.
Reply: We appreciate your comment. This part is not fully explained. We add more sentences This part is not fully explained. We add more sentences asas ””The diabetic retinopathy can be classified into background diabetic retinopathy The diabetic retinopathy can be classified into background diabetic retinopathy (BDR), pre(BDR), pre--proliferative diabetic retinopathy (preproliferative diabetic retinopathy (pre--PDR), and proliferative diabetic PDR), and proliferative diabetic retinopathy (PDR). BDR presentsretinopathy (PDR). BDR presents dotdot--blot or flameblot or flame--shaped retinal hemorrhages, hard shaped retinal hemorrhages, hard exudates and microaneurysm in ophthalmoscope. PDR presents same features in exudates and microaneurysm in ophthalmoscope. PDR presents same features in ophthalmoscope of BDR plus neovascularization in optic disc or retina. Preophthalmoscope of BDR plus neovascularization in optic disc or retina. Pre--PDR is a PDR is a stage between BDR and PDR, and it has same fstage between BDR and PDR, and it has same features in ophthalmoscope of BDR plus eatures in ophthalmoscope of BDR plus retinal nerve fiber layer infarction (seen as cottonretinal nerve fiber layer infarction (seen as cotton--wool spot in ophthalmoscope), but wool spot in ophthalmoscope), but no neovascularization in optic disc or retina.” in the “Clinical Features and Stages of no neovascularization in optic disc or retina.” in the “Clinical Features and Stages of Diabetic Retinopathy” section.Diabetic Retinopathy” section.
4. Age, seAge, sex, diopter, glaucoma and other retinal disease may affect retinal oxygen vessel x, diopter, glaucoma and other retinal disease may affect retinal oxygen vessel saturation. Does the four groups in the present study control these variablessaturation. Does the four groups in the present study control these variables.
Reply: We appreciate your comment. IIn this study, we only present the concept about n this study, we only present the concept about information information of of retinal oxygen vessel saturation by our method. We analysis these retinal oxygen vessel saturation by our method. We analysis these ophthalmoscope imagesophthalmoscope images based on four DR stages without control age, sex, diopter, based on four DR stages without control age, sex, diopter, glaucoma and other retinal disease. This is an interesting research area. We will do this glaucoma and other retinal disease. This is an interesting research area. We will do this part follow yourpart follow your comments in the future.comments in the future.

Round 2

Reviewer 1 Report

Please accept the manuscript at the present form.

Author Response

Reviewer 1:

Please accept the manuscript at the present form.

Reply:

We appreciate your comment.

Reviewer 2 Report

To adequately describe the classification of DR, Please add the reference “Bowling, B. Kanski's clinical ophthalmology: a systematic approach; Saunders Ltd: 2015” in chapter 14: Retinal Vascular Disease in Section 2.3 Clinical Features and Stages of Diabetic Retinopathy. (Line 128, Page 3)

Author Response

Reviewer 2:

To adequately describe the classification of DR, Please add the reference “Bowling, B. Kanski's clinical ophthalmology: a systematic approach; Saunders Ltd: 2015” in chapter 14: Retinal Vascular Disease in Section 2.3 Clinical Features and Stages of Diabetic Retinopathy. (Line 128, Page 3)

Reply:

We add the reference as [24] in Line 128, Page 3.